# Mixture of Sparse Attention: Content-Based Learnable Sparse Attention via MoEs

## Abstract

Recent advances in large language models highlighted the excessive quadratic cost of self-attention. Despite the significant research efforts, subquadratic attention methods still suffer from inferior performance in practice. We hypothesize that dynamic, learned content-based sparsity can lead to more efficient attention mechanisms. We present Mixture of Sparse Attention (MoSA), a novel approach inspired by Mixture of Experts (MoE) with expert choice routing. MoSA dynamically selects tokens for each attention head, allowing arbitrary sparse attention patterns. By selecting $k$ tokens from a sequence of length $T$, MoSA reduces the computational complexity of each attention head from $O(T^2)$ to $O(k^2 + T)$. This enables using more heads within the same computational budget, allowing higher specialization. We show that among the tested sparse attention variants, MoSA is the only one that can outperform the dense baseline, sometimes with up to 27% better perplexity for an identical compute budget. MoSA can also reduce the resource usage compared to dense self-attention. Despite using torch implementation without an optimized kernel, perplexity-matched MoSA models are simultaneously faster in wall-clock time, require less memory for training, and drastically reduce the size of the KV-cache compared to the dense transformer baselines.

## 1 Introduction

Modern transformer architectures (Vaswani et al., 2017) have proven to be highly effective for sequence modeling tasks and are the key to the success of large language models (LLMs; (Brown et al., 2020; Touvron et al., 2023; Team et al., 2024; Grattafiori et al., 2024)). One of the key components of their success is the attention mechanism, which enables dynamic information propagation by computing weighted sums of past states for each token. This results in high computational and memory complexity, both quadratic in sequence length. The key to the success of LLMs is the ever-increasing model sizes and context windows. Training and deploying these models becomes increasingly prohibitive. Furthermore, the KV-cache memory footprint during inference presents a significant bottleneck, limiting practical deployment and increasing operational costs.

This led the researchers to explore alternative approaches. State Space Models (Gu et al., 2020; 2022; Gu and Dao, 2023; Wang et al., 2024; Yang et al., 2025) capture long-range dependencies with just a handful of state variables rather than relying on full attention matrices. They, however, fall short of full self-attention in terms of practical performance. To counteract lossy compression of State Space Models, a recent line of work investigates hybrids that combine quadratic attention and linearized memories (Park et al., 2024; Zuo et al., 2022; Lieber et al., 2024). Linear attention (Katharopoulos et al., 2020; Schlag et al., 2021; Schmidhuber, 1992)[*] optimizes the attention cost by changing the order of the operations in the attention after removing nonlinearity. However, it also performs poorly compared to quadratic attention (Qin et al., 2022).

As an alternative, static sparse attention methods (Child et al., 2019) reduce the quadratic complexity by selectively attending to a subset of tokens to be used in the attention. They use hand-defined coarse-grained patterns that are not data-dependent. Typical examples of these methods are the block-sparse and strided attention (Child et al., 2019; Zaheer et al., 2020; Beltagy et al., 2020). Static sparsity and block aggregation methods, however, impose significant limitations. They encourage the

---

[*]Note that *unnormalized linear transformers* (with "linear attention") were first published in 1992 under the name *fast weight controllers* (Schmidhuber, 1992) or *fast weight programmers*.

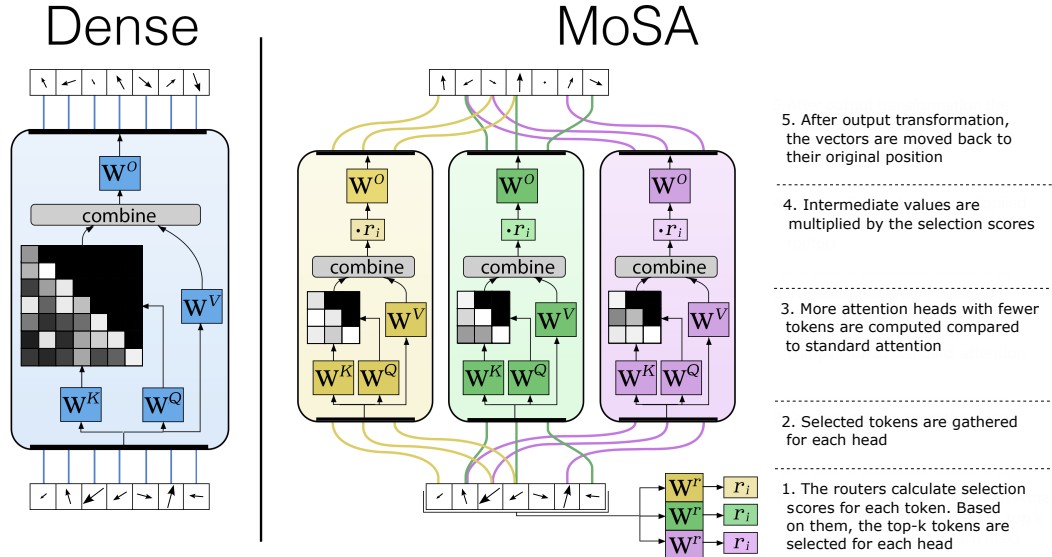

Figure 1: MoSA layer compared to the dense attention layer. MoSA replaces each dense head with multiple heads with a learnable sparsity pattern. Each head selects its own $k$ tokens to process. MoSA calculates query, key, and value projections only for the selected token and computes the attention only between them. It drops the rest of the tokens, leading to more efficient compute utilization. This reduces the computational and memory complexity on a sequence of length $T$ from $O(T^2)$ to $O(k^2 + T)$. The saved compute budget can be used to scale up the number of heads.

compression of multiple tokens into a single, lossy representation. This is necessary to remember information beyond the active block. Such compression makes fine-grained recall difficult. The problem is similar to the well-known limitation of state-space models, which are forced to compress the entire past into a fixed-size representation (Arora et al., 2024; Jelassi et al., 2024). Content-based dynamic sparse attention (Tay et al., 2021; Vyas et al., 2020; Roy et al., 2021) methods can, in principle, learn to attend to individual tokens, regardless of their location in the input, while ignoring less useful tokens. The Routing Transformer (Roy et al., 2021) clusters the tokens within each head using online K-means. However, it fails to show significant performance gains over static sparse-attention methods, possibly due to the slow convergence of online K-means (Bottou and Bengio, 1994).

We propose a novel approach, inspired by Mixture-of-Experts (Shazeer et al., 2017; Fedus et al., 2022; Hampshire and Waibel, 1989; Jacobs et al., 1991), to create a dynamic, content-based, and head-specific selection of tokens for sparse attention. This is achieved with Expert-Choice Routing (Zhou et al., 2022), where each attention head is treated like an expert and selects its own specific tokens from the input. This creates a perfectly balanced selection, avoiding the need for complicated regularization techniques. We name our approach Mixture-of-Sparse Attention(*MoSA*). Although recent work explored applying ideas from the MoE literature to attention mechanisms (Zhang et al., 2022; Csordás et al., 2024), they focus on reducing the number of materialized attention matrices. We propose a different approach: we make the attention matrices sparse by selecting a small subset of tokens for each attention head.

By selecting $k$ tokens from a sequence of length $T$, MoSA reduces the computational complexity of the attention head from $O(T^2)$ to $O(k^2 + T)$. Sparse attention techniques have historically been employed out of necessity to manage long sequences that exceed available computational capacities. In contrast, we also explore the use of the saved computation budget for creating additional attention heads. Thus, in this setup, MoSA employs a large number of highly sparse attention heads, encouraging their specialization. We show that this allows for better utilization of the available compute budget and leads to substantially better iso-flop language modeling performance compared to dense attention. Furthermore, we analyze other sparse attention methods, such as fixed sparse attention (Child et al., 2019) and the Routing Attention (the attention introduced in the

Routing Transformer) (Roy et al., 2021). MoSA is the only sparse attention method we analyzed that demonstrates improvement over dense baselines in the IsoFLOP setting.

Following observations of sparse attention methods (Child et al., 2019; Roy et al., 2021), we incorporate MoSA as part of a hybrid model with a different type of attention. Our main results demonstrate that hybrid models with many MoSA and four dense heads significantly improve the model's quality by up to 27% in an IsoFLOP setting. Specifically, we evaluate MoSA on a language modeling task by starting with dense baselines and incrementally sparsifying the attention. We ensure FLOP-matching by swapping a specific number of dense heads for *more* sparse heads. We repeat this procedure on different scales, starting with baselines from 28M to 516M parameters. MoSA consistently improves perplexity across all model scales. Furthermore, we demonstrate that for long sequences, MoSA combined with local attention heads clearly outperforms other analyzed sparse attention methods with a fixed budget.

The IsoFLOP results demonstrate MoSA's superior performance in a FLOP-matched setting. However, sparse attention methods are often used to reduce computational and memory requirements. Furthermore, the idealized FLOP requirements often do not reflect wall-clock time. To demonstrate MoSA's efficiency, In Section 3, we show that in a perplexity-matched setting, MoSA exhibits both improved wall-clock time and GPU memory consumption even without a specialized CUDA kernel. It also reduces the total number of keys and values used in the computation, resulting in a significantly smaller KV cache. KV-cache size is an important practical problem for LLM inference and is the main focus of many post-training sparse attention methods (Li et al., 2025; Cai et al., 2024).

In summary, our contributions are the following: 1. We propose MoSA, a sparse attention method that uses a *learned*, context-based token selection, with each of the heads attending to a small subset of all tokens. 2. We evaluate MoSA in an IsoFLOP setting on four different scales with dense baselines ranging from 28M parameters to 516M. In this setting, MoSA improves perplexity by up to 27%. MoSA is the only sparse attention method we analyzed that improved perplexity compared to the dense baseline. 3. We demonstrate that, in a perplexity-matched setting, a pure PyTorch implementation of MoSA improves both wall-clock time and memory usage simultaneously, without requiring specialized fast kernels. This setup also drastically reduces the KV cache size by using only a small subset of keys and values. 4. We demonstrate that on long sequences, MoSA maintains a large advantage compared to other tested sparse-attention methods. 5. For completeness, we experiment with a variant of MoSA, called *MoSAIC*, based on token-choice routing.

## 2 METHOD

### 2.1 BACKGROUND

**Attention Mechanism.** Attention assigns input-dependent weights to tokens in a sequence, allowing each token to gather context from the rest of the sequence. To do this, each token is projected to three vectors: its *query*, *key*, and *value*. For a given token, we compare its *query* vector with the *key* vectors of all tokens (including itself), producing a set of similarity scores. The scores are then normalized and used to calculate a weighted sum of the tokens' *value* vectors. The result is a new representation that dynamically integrates information throughout the sequence.

Let $T$ be the sequence length, $h$ the hidden dimension of the model, and $h'$ the hidden dimension in each head. $Q, K, V \in \mathbb{R}^{T \times h'}$ represents the query, key and value matrices, respectively.

The attention is computed as:

$$\text{Attention}(Q, K, V, M) = \text{softmax}\left(\frac{QK^\top + M}{\sqrt{h'}}\right) V \tag{1}$$

Here, $M$ denotes the attention mask that represents hard modeling constraints. $M_{i,j} = 0$ if and only if $i'th$ token is allowed to attend to $j'th$ token, otherwise $M_{i,j} = -\infty$. In causal language models, $M_{i,j} = 0 \iff i \geq j$ ensures that no token can attend to the future.

The multi-head attention (MHA) creates multiple instances (heads) of query, key, and value matrices from an input sequence $X \in \mathbb{R}^{T \times h}$ and applies the attention to each instance independently. Each

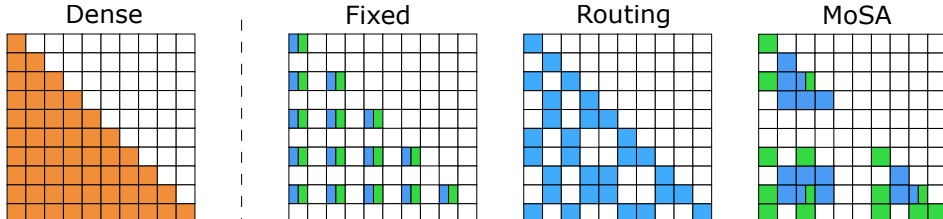

Figure 2: Attention variants visualized. In the plot, the colors indicate different heads. Sparse attention methods are roughly FLOP-matched and have sparsity $\rho = 2$. One Routing Attention head corresponds in FLOP-cost to $\rho$ Fixed/MoSA heads. Fixed sparse attention uses only $k = \frac{T}{\rho}$ tokens in specific positions, with regular stride. The Routing Attention clusters tokens within each head into $\rho$ clusters of size $k$ based on their representations. MoSA selects $k$ tokens for each attention head independently based on their representations.

head has its own mappings $W_i^Q, W_i^K, W_i^V \in \mathbb{R}^{h \times h'}$ and $W_i^O \in \mathbb{R}^{h' \times h}$, where $i \in \{1..H\}$ and $H$ is the number of heads. $h'$ is typically set to $\frac{h}{H}$. $Q_i = XW_i^Q, K_i = XW_i^K, V_i = XW_i^V$.

$$X_{out} = \sum_{i=1}^{H} \text{Attention}(Q_i, K_i, V_i, M)W_i^O \tag{2}$$

The resulting mechanism allows the model to adaptively focus on relevant information while maintaining differentiability. The lack of recurrence in the operations enables parallel processing of sequence elements. However, $\mathbf{QK}^\top$ is a $T \times T$ matrix and therefore introduces quadratic computational and memory complexity as a function of the sequence length.

**Mixture of Experts.** Mixture of Experts (MoE) combines multiple specialized neural networks (experts) with a gating mechanism that learns to route each input to the best-matching experts, activating only a small subset of experts per example. An MoE layer then computes its output as a sparsely weighted combination of the predictions of selected experts, with routing weights dynamically determined by the gating network.

Formally, given an input $\boldsymbol{x} \in \mathbb{R}^h$, the MoE layer with $E$ experts and a scoring function (a router) $sel : \mathbb{R}^h \to \mathbb{R}^n$ can be expressed as $y(\boldsymbol{x}) = \sum_{i \in \mathcal{E}} r_i(\boldsymbol{x})E_i(\boldsymbol{x})$ where $y(\boldsymbol{x})$ is the final output of the layer and $E_i(\boldsymbol{x})$ is the output of the expert $i$. $\mathcal{E}$ is the set of selected experts, usually defined as $\mathcal{E} = \text{argtopk}(r(\boldsymbol{x}) + \varepsilon, k)$, where $k \in \mathbb{N}$ is the number of active experts, $\varepsilon$ is a stochastic noise present only during the training for exploration. The inputs are processed only by the active experts.

In contrast, Expert-Choice routing (Zhou et al., 2022) ensures perfect load balancing by inverting the traditional routing paradigm. Instead of the tokens choosing their experts, the experts choose which inputs they prefer to process. Given a batch of $B$ inputs, each expert selects the top-$k$ out of the $B$ inputs it will process.[†]

## 2.2 MIXTURE OF SPARSE ATTENTION (MOSA)

Sparse attention methods model global dependencies by selecting specific tokens that can attend to other specific tokens based on a hand-engineered set of rules (Beltagy et al., 2020; Zaheer et al., 2020) or by blockwise aggregation of tokens (Yuan et al., 2025). Both of these families of methods impose the mixing of information during token aggregation, either explicitly or implicitly.

We propose instead to select tokens adaptively for each head based on the input. Thus, a flexible set of important tokens can be kept around, creating content-based sparsity without the need for information mixing. To achieve that, we take inspiration from Expert-Choice routing in MoEs. We name our method *Mixture of Sparse Attention (MoSA)*. MoSA learns which individual tokens to use for attention through end-to-end training. Each attention head in MoSA learns its own unique sparsity

---

[†]In our case, each expert selects top-$k$ tokens from the sentence to process independently for each batch.

pattern, allowing different heads to specialize in different subsets of tokens relevant to their particular function within the network. This diverse, head-specific token selection pattern ensures that the model preserves the granular information within each relevant token while dynamically discovering optimal sparsity patterns specific to the data distribution. The architectural difference between MoSA and dense attention is illustrated in Fig. 1.

| Model size | #Params Dense | Dense ppl ↓ | MoSA Best ppl ↓ | Fixed Best ppl ↓ | Routing Best ppl ↓ |
|---|---|---|---|---|---|
| Tiny | 28M | 22.46 | 16.37 (−27.1%) | 23.28 (+3.7%) | 23.33 (+3.9%) |
| Small | 113M | 16.01 | 12.97 (−19.0%) | 16.51 (+3.1%) | 16.43 (+2.6%) |
| Medium | 210M | 13.95 | 11.22 (−19.6%) | 14.35 (+2.9%) | 14.21 (+1.9%) |
| Large | 516M | 12.20 | 11.15 (−8.6%) | 12.40 (+1.6%) | 12.24 (+0.3%) |

Table 1: Comparing dense and sparse models (Fixed, Routing, MoSA) under a fixed computational budget (see Section 3). For sparse models, the table contains the best perplexity across all sparsities bigger than 1. The results for sparse models were selected as the best of all sparsities. Relative difference to the dense baseline is displayed in the parentheses. MoSA significantly outperforms the dense baseline, reducing perplexity by up to 27%. The fixed and the Routing Transformer baselines both fail to reach the performance of the dense model.

The sparsity in MoSA reduces the computational cost of each attention head, allowing the use of more heads to develop targeted projections optimized for specific relationship types. The computational savings are particularly substantial when the number of selected tokens is significantly smaller than the sequence length.

In MoSA, in addition to the standard projections, each head has an additional router that selects which tokens are used for that head. Formally, the router is defined using the weight matrix $W^r \in \mathbb{R}^h$. Let $X \in \mathbb{R}^{T \times h}$ be the $T$-long sequence of input tokens. The router calculates the selection scores for each token $r = \sigma(XW^r) \in \mathbb{R}^T$. For $\sigma$ we use the non-competitive sigmoid function $\sigma(x) = \frac{1}{1+e^{-x}}$ following observations from $\sigma$-MoE (Csordás et al., 2023). Subsequently, we use expert choice for the selection of tokens for each head:

$$r^{topk}, I = TopK(r, k)$$

where $TopK$ returns the highest $k$ values of $r$ called $r^{topk} \in \mathbb{R}^k$, along with their indices $I \in \{0, ..., T-1\}^k$. $I$ is used to select the subset of inputs for the MoSA head:

$$X^s = (X_{I_1}, X_{I_2}, ..., X_{I_k}) \in \mathbb{R}^{k \times h}$$

where $X_i$ represents $i'th$ row from matrix $X$. After that, queries, keys, and values are calculated identically to the standard MHA: $X^s$ as $Q = X^s W^Q, K = X^s W^K, V = X^s W^V$. As our primary target is language modeling, we also calculate the mask that prohibits attending to future tokens. Unlike the standard MHA, this mask is not triangular and has to take into account the token indices selected by the head: $M_{i,j} = 0 \iff I_i \geq I_j, -\infty$ otherwise.

The sparse attention can be computed using the standard attention defined in Eq. 1. $A = \text{Attention}(Q, K, V, M)$. This allows the combination of MoSA with optimized attention implementations such as Flash Attention (Dao et al., 2022). The resulting vectors $A_i$ are multiplied by the corresponding router values $r_i$. Then, after the output transformation $W^o$, they are moved back to their original positions in the full-length sequence $Y \in \mathbb{R}^{T \times h}$.

$$X^o = \text{diag}(r)AW^o \in \mathbb{R}^{k \times h}$$

$$Y_j = \begin{cases} X_i^o, & \text{if } j = I_i \text{ for some } i \in \{1, \dots, k\}, \\ 0, & \text{otherwise,} \end{cases} \quad \text{for } j = 1, \dots, T.$$

$\text{diag}(\cdot)$ creates a diagonal matrix from a vector, used for elementwise scaling of the columns of the matrix $A$ by a vector $r$. This ensures that the token's contribution is proportional to the router's output. This also enables the router to receive gradients, making it learnable by gradient descent.

We call the combined transformation of $x$ into $y$, parameterized by $\theta_i = (W^Q, W^K, W^V, W^O, W^r)$ a single MoSA head: $Y = \text{MoSA}_{head}(X; \theta_i)$. A MoSA layer parameterized by $\theta = \{\theta_i\}_{i \in 1...H}$ is a sum of all MoSA heads

$$\text{MoSA}(X; \theta) = \sum_{i=1}^{H} \text{MoSA}_{head}(X; \theta_i) \tag{3}$$

The entire transformation in the multihead version can be efficiently implemented in PyTorch (Paszke et al., 2019) using `einsum`, `scatter` and `gather` operations.

**Hybridization.**    Sparse attention methods are usually combined with local attention (Child et al., 2019; Roy et al., 2021) when used on long sequences. Sparse attention then captures global dependencies, while local attention preserves local context. As our setup permits the use of dense attention, in our main experiments, we combine MoSA or corresponding sparse attention baseline with 4 dense heads. In Appendix D, we demonstrate the necessity of hybridization and motivate our selection of four dense heads for the models. In Section 3, we combine MoSA with local attention for long sequences and demonstrate that MoSA demonstrates superior performance in this scenario as well.

**MoSAIC.**    Building on the central idea of MoSA i.e., learned per-head content-based token selection, we further explore how this principle can be adapted to the native autoregressive generation. To this end, we introduce *Mixture of Sparse Attention with Independent Choice (MoSAIC)*, a variant of MoSA that is natively autoregressive. MoSAIC replaces MoSA's expert-choice procedure with standard token-choice routing so that each token independently selects the attention heads it will use, thereby natively preserving causal decoding. The full details of the routing algorithm and extensive analysis of design variants is available in Appendix E.

# 3 EXPERIMENTS

In this section, we empirically demonstrate MoSA's performance in different settings. We compare MoSA to dense and sparse baselines. In Section 3, we evaluate all the methods on language modeling under a fixed FLOP budget. In Section 3 we demonstrate the practical benefits of MoSA by measuring wall-clock time, memory usage, and KV cache size in a perplexity-matched setup. In Section 3 we investigate the performance of MoSA on long sequences. Furthermore, in Appendix 3, we analyze the performance of different models in downstream zero-shot tasks.

We use four model sizes: *Tiny*, *Small*, *Medium* and *Large*. Each size is defined by the *FLOP count* of the forward pass of the corresponding dense transformer baseline. The parameter count of dense models associated with each size is: 28M for *Tiny*, 113M for *Small*, 210M for *Medium*, and 516M for *Large*.

Apart from a dense baseline, we compare MoSA with two sparse attention methods: Fixed sparse attention with strided selected tokens that participate in the attention and Routing Attention that represents a content-based sparse attention baseline. Different attention methods are visualized in Fig. 2. The baselines are described in detail and compared with MoSA in App. A.

**IsoFLOP Curves**    We evaluate sparse methods by gradually increasing sparsity rate $\rho = \frac{T}{k}$. This reduces the compute requirements for each sparse head. We use the saved budget to increase the number of sparse heads. Specifically, we choose the number of sparse heads to be the maximum such that the FLOPs of the sparse model do not exceed the FLOPs of the baseline model for a given size. All sparse models include four dense heads that we keep (see Section 2.2), and are included in the FLOP calculations. Note that increasing the number of attention heads also increases the memory requirements of all methods. Consequently, for the larger FLOP-matched models, we restricted the explored sparsity values to ensure that the models fit in the memory budget dictated by our hardware.

Starting from sparsity 1, which corresponds to the dense model, we gradually increase the sparsity and measure the test-set perplexity of FLOP-matched models. Table 1 lists the best results for each model class and size. Across all model sizes tested, MoSA achieved significantly better perplexity within fixed FLOP budgets compared to dense baselines. All MoSA hybrids reduce the perplexity of the baseline, sometimes by $27\%$. On the other hand, the sparse baselines for all sparsities $\rho > 1$ perform worse than the dense baseline.

Figure 3 illustrates the IsoFLOP curves of the models with varying degrees of sparsity. For MoSA, performance steadily improves as sparsity increases, reaching optimal results at approximately $\rho = 64$. Beyond this threshold, performance begins to decline, creating a "U" shape in the curve. This is likely because the excessively high sparsity values limit the model's ability to capture complex attention patterns. For example, at $\rho = 256$ with a sequence length of $T = 1024$, only $k = 4$ tokens are selected to participate in each attention head.

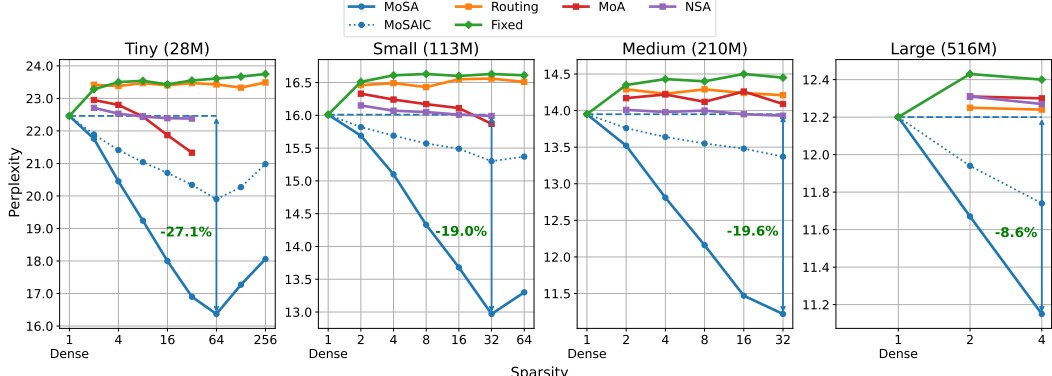

Figure 3: Perplexity (↓) of FLOP matched models under different sparsities. Each plot corresponds to a specified FLOP budget per step. The number in parenthesis is the number of parameters of the dense baseline. Sparsity 1 represents the dense baseline. As sparsity increases, MoSA's perplexity improves monotonically until reaching a saturation point around sparsity 32-64, beyond which performance deteriorates. This is likely because at very high sparsity levels, each attention head selects only a few tokens, which is insufficient to capture the complex relations. MoSAIC, the token-choice based version of MoSA also leads to significant improvements in comparison to dense model. On the other hand, other sparse methods fail to reach the perplexity of the dense baseline in the IsoFLOP setting. We explore fewer sparsity levels for larger models due to excessive memory requirements.

For some configurations, MoSA turns proves to be more efficient than the dense model even in a parameter-matched setting. For example, *Medium* model with sparsity 8 has $442M$ parameters and perplexity 12.16, while the *Large* baseline model has $516M$ parameters and perplexity 12.20. This shows that a higher specialization of the heads might lead to improved performance even when we discard computational benefits. Detailed results for different MoSA sparsity configurations, together with the total number of parameters and the number of heads, are listed in the Appendix 5.

In contrast to MoSA, both fixed sparse attention and the Routing Attention consistently underperform the dense baseline across all sparsity levels. They exhibit relatively constant, but worse, perplexity across different sparsity values, with only minor fluctuations that reveal no discernible trend.

**Resource Optimization**  The previous section demonstrates MoSA's ability to achieve better perplexity than dense transformers with an identical compute budget. In this section, we examine MoSA's practical efficiency gains. Specifically, we match the perplexity scores between the MoSA and the dense baseline to measure wall-clock time, memory, and KV-cache size savings.

To find the perplexity-matched comparison, we select sparsity to be equal to 32 for model sizes *Tiny, Small* and *Medium*. For *Large* we select $\rho = 16$ to keep sparsity closer to the range investigated in Section 3. Then, we gradually increase the number of MoSA heads until the perplexity matches the dense baseline. We do it for all four model scales defined in Section 3.

The results are shown in Table 2. MoSA can match the dense baseline, while being faster in wall-clock time and using less memory at the same time. These findings show that MoSA not only improves model quality in the FLOP-matched setting but can also be used to reduce computational and memory requirements when targeting the same performance level. Furthermore, it shows that MoSA uses computation more effectively than standard dense attention across all efficiency metrics. MoSA achieves this without a specialized CUDA kernel using only PyTorch-level operations. We expect that designing a specialized kernel would result in additional significant efficiency gains.

In addition to the speed and memory used for the training, we report the total number of key-value pairs (KV) used, calculated as $\text{KV} = T H_{dense} + k H_{mosa}$, where $H_{dense}$ and $H_{mosa}$ represent the number of dense and sparse heads, respectively. KV directly corresponds to the size of the costly KV-Cache in the autoregressive setting. KV cache optimization has been the goal of many post-training sparse-attention methods(Liu et al., 2023; Li et al., 2025; Cai et al., 2024). Our results demonstrate

that MoSA offers a significant reduction in KV-cache size while simultaneously improving speed and memory requirements.

| | Tiny | | Small | | Medium | | Large | |
|---|---|---|---|---|---|---|---|---|
| | Dense | MoSA | Dense | MoSA | Dense | MoSA | Dense | MoSA |
| Dense Heads | 9 | 4 | 9 | 4 | 9 | 4 | 16 | 4 |
| MoSA Heads | 0 | 17 | 0 | 14 | 0 | 12 | 0 | 16 |
| Perplexity ($\downarrow$) | 22.46 | 22.40 | 16.02 | 16.01 | 13.94 | 13.76 | 12.20 | 12.16 |
| Wall-time/step $\downarrow$ (ms) | 137 | **127** | 326 | **319** | 619 | **592** | 807 | **703** |
| Wall-time/step gain (%) | – | $-7.3\%$ | – | $-2.1\%$ | – | $-4.4\%$ | – | $-12.9\%$ |
| Memory $\downarrow$ (GB) | 21.1 | **19.0** | 32.4 | **31.4** | 50.2 | **49.4** | 104.1 | **94.5** |
| Memory gain (%) | – | $-10.0\%$ | – | $-3.1\%$ | – | $-1.6\%$ | – | $-9.2\%$ |
| KV Total $\downarrow$ (K) | 9.2 | **4.5** | 9.2 | **4.4** | 9.2 | **4.4** | 16.4 | **5.0** |
| KV Total gain (%) | – | $-51.1\%$ | – | $-52.2\%$ | – | $-52.2\%$ | – | $-69.5\%$ |

Table 2: Resource usage reduction from perplexity-matched MoSA models. KV is the KV-cache size, representing the total number of key-value pairs required (in thousands). MoSA models match the perplexity of dense baselines while at the same time improving wall-clock time, using less memory, and significantly smaller KV cache for all model sizes. Resource usage was measured on a single A100 GPU for *Tiny, Small* and *Medium* models and on two A100 GPUs for *Large*.

**Scaling with Sequence Length** Traditionally, sparse attention methods have been introduced as a necessity when sequence length makes dense attention computationally prohibitive. After demonstrating MoSA's effectiveness in standard-length sequences, we now investigate MoSA's benefits in this long sequence setup.

In contrast to previous sections, here we combine MoSA or a baseline method with local attention (Child et al., 2019; Beltagy et al., 2020). We use local attention instead of dense attention because even a small number of dense attention heads would result in prohibitive memory usage in a longer context scenario. This is a standard practice in the sparse attention literature (Child et al., 2019; Roy et al., 2021). Local attention preserves local dependencies, while global, sparse attention enables efficient processing of long dependencies.

We scale our sequence length from 1024 to 8192 tokens and keep the $k$ constant equal to 64. Hence, the sparsity increases from $\rho = 16$ for $T = 1024$ to $\rho = 128$ for $T = 8192$. Contemporary sparse attention methods for long sequences are trained in longer sequences (Yuan et al., 2025). However, due to our limited hardware budget, we restrict our experiments to a sequence length of 8192. We treat this investigation as a preliminary analysis that demonstrates the potential of MoSA for long sequences. Importantly, it demonstrates that MoSA performs well when combined with local attention, which is a typical long-sequence setup.

As in the previous section, we compare MoSA with fixed sparse attention and the Routing Attention. All long sequence models have 6 layers and hidden dimension size of 1024. The Routing Transformer has 4 local attention heads and 4 Routing Transformer heads in all layers, whereas the fixed sparse attention and MoSA have 60 sparse heads and 4 local attention heads. We chose 60 sparse heads to roughly FLOP match all models for $T = 1024$. However, as we keep $k$ constant, for longer sequences with 2048, 4096 and 8192 tokens, the FLOP cost for fixed attention and MoSA will be much lower than for the Routing Attention. For $T = 8192$ FLOP cost of 60 MoSA's heads is equal to only 22.99% of 4 Routing Transformer heads.

The results are shown in Fig. 4. MoSA significantly outperforms other sparse attention methods across all sequence lengths. This is true even at length 8192, where MoSA uses only a small fraction of the computational cost of the Routing Transformer. The significant performance gap in the results demonstrates the potential of MoSA for ultra-long sequences (Kitaev et al., 2020; Yuan et al., 2025; Xu et al., 2025a). Given our limited resources, we leave the investigation of MoSA in this context for future work.

## 4 LIMITATIONS AND FUTURE WORK

As in other context-based sparse attention methods, the perplexity gains do not always translate to downstream task performance (App. F). This discrepancy stems from two distinct factors: First,

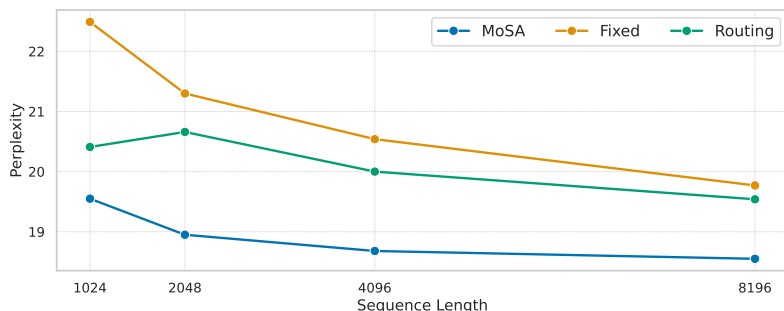

Figure 4: Perplexity of sparse-attention methods (MoSA, Fixed, and Routing) as sequence length increases. Each method has a fixed size window size (cluster size for the Routing Transformer, number of tokens selected for each head in MoSA and Fixed) regardless of total sequence length. MoSA matches the computational cost of the fixed sparsity baseline while requiring fewer FLOPs than the Routing Attention and consistently achieves the lowest perplexity.

sparse attention methods generally underperform on tasks consisting of short sequence lengths. Practitioners have shown that additional training with truncated sequences might alleviate this problem. Second, MoE architectures experience performance gaps in downstream tasks despite strong language modeling capabilities, although recent research demonstrates that instruction tuning can help significantly (Shen et al., 2024a). We consider exploring methods to mitigate the discrepancy between perplexity and downstream task performance in future work; MoSAIC already demonstrates promising progress in this direction.

Several promising research directions emerge from this work. Further exploration of MoSA's effectiveness on longer sequences. Furthermore, combining multiple sparse attention methods often leads to synergic improvements on long sequences (Zaheer et al., 2020; Beltagy et al., 2020). Thus, we expect that combining other sparse head types with MoSA could lead to additional benefits. From an implementation perspective, developing specialized CUDA kernels would further improve efficiency. MoSA could be integrated with complementary approaches such as MQA(Shazeer, 2019), GQA(Ainslie et al., 2023), or SwitchHead(Csordás et al., 2024) to improve the efficiency even further. Furthermore, exploring MoSA on other modalities, particularly vision transformers, could yield valuable insights into the method's versatility across different data types and architectures.

## 5 CONCLUSIONS

This paper introduces Mixture of Sparse Attention (MoSA), a novel attention architecture that selectively focuses on the most relevant tokens for the attention head, redirecting saved compute to create additional heads. MoSA reduces the computational complexity of attention from $O(T^2)$ to $O(k^2 + T)$, where $T$ is the sequence length and $k$ is the number of selected tokens per head.

Unlike other sparse attention methods that primarily show benefits for extremely long sequences, MoSA delivers substantial performance gains even in standard-length contexts. MoSA significantly outperforms both dense attention and sparse methods like fixed attention or the Routing Transformer, achieving up to 27% perplexity improvement over dense baselines across models of different scales. MoSA can also be used to reduce the resource requirements of the models, including a more than $50\%$ reduction in the KV-cache size. Additionally, our results indicate that MoSA maintains its superiority in long-sequence scenarios, outperforming other sparse attention methods in these contexts as well.

The efficiency and corresponding performance gains demonstrated by MoSA have significant implications for the design of adaptive architectures. MoSA or subsequent adaptive models stemming from MoSA can be used for reducing the training costs and environmental impact of large language models, potentially enabling more economical scaling while lowering energy consumption and carbon emissions. Given its versatility and performance advantages, we anticipate that MoSA will drive innovations in both transformer architecture research and industrial applications.

## REPRODUCIBILITY STATEMENT

We provide our source code to reproduce the results from experimental section in the supplementary material. Furthermore, in the appendix we include the details of each model together with parameters of the training runs and machines used for the experiments.

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

# A BASELINES

Apart from a dense baseline, we compare MoSA with two sparse attention methods: static, position-based sparse attention, and content-based sparse attention.

**Fixed Sparse Attention.** Position-based static attention patterns have been shown to be a strong sparse attention variant (Child et al., 2019), outperforming strided sliding window attention. Fixed sparse attention for a sparsity $\rho$ selects $k = \frac{T}{\rho}$ tokens with stride $\rho$. Using the notation introduced in Section 2.2, fixed sparse attention can be written as a special case of MoSA, where $I = [0, \rho, 2\rho, ..., T - \rho]$ and $r = 1$.

Fixed sparse attention reduces computational complexity in two ways. First, it decreases the $O(T^2)$ cost of the full attention matrix by limiting attention to predefined token positions. Second, since only these pre-selected tokens participate in attention calculations, the query, key, value, and output transformations need only be computed for this subset rather than all tokens. This is important because MoSA also benefits from calculating transformations only for a selected subset of tokens. Hence, investigating this fixed sparse attention gives insight on whether pure benefits of sparsifying over transformations can lead to performance improvements.

However, fixed sparse attention introduces information flow constraints. Pre-selected tokens must aggregate necessary information in earlier layers. Furthermore, in the subsequent layers they have to be routed back to the positions where they are most useful. This additional overhead in information routing limits the model's representational capacity and overall expressiveness.

**The Routing Transformer.** We also compare MoSA to the content-based attention proposed in the Routing Transformer (Roy et al., 2021). The Routing Attention is the most similar method to MoSA we found in the literature. It groups tokens with online K-means into $\rho$ clusters of size $k = \frac{T}{\rho}$ *inside* each head. This is implemented during training by the top-k tokens most similar to the cluster centers using the dot-product distance metric. Cluster centers are learned using a moving average of the most similar tokens.

The Routing Attention might resemble the Expert-Choice selection with MoSA. There are, however, several crucial differences that, as our experiments show, lead to significant differences in the performance of MoSA in comparison to the Routing Transformer. Specifically, online K-means, used for clustering in the Routing Transformer is known for suffering from an extremely slow convergence rate (Bottou and Bengio, 1994). It is also unclear if clustering keys and queries is well aligned with the language modeling objective. In contrast, the learned dynamic matching mechanism of MoSA is directly optimized by the same objective as the model.

MoSA benefits from the sparsity in the $W^Q, W^K, W^V, W^O$ transformations, which need to be computed only for selected tokens. In contrast, the Routing Transformer has to compute all keys and queries before the clustering step. MoSA's efficiency enables the use of more heads with specialized weights in a smaller subset of tokens. Its selection can also lead to dynamic compute allocation, where some more important tokens are processed by more heads than less important tokens.

Last but not least, the Routing Transformer performs best in language modeling when the clusters share the same destination (query) tokens and source (keys and values) tokens. In our experiments, we also found that MoSA performs better if the same tokens are selected for the source and destination sides. However, to enforce this in the Routing Transformer, they require to set $W^Q = W^K$. In MoSA, however, the same selection for source and destination side can be enforced with $W^Q$ different from $W^K$, allowing greater flexibility.

We visualize typical schematic attention patterns of the baselines and MoSA in Fig. 2. Note that several previous works proposed combining different types of sparse attention to achieve synergic performance in long-sequence tasks (Beltagy et al., 2020; Zaheer et al., 2020; Zhang et al., 2023). In this work, we focus on investigating sparse attention methods in combination with a few dense attention heads, but without combining multiple sparse attention types. We leave combining MoSA with other sparse-attention methods for future work.

**Mixture of Attention Heads** Mixture of Attention Heads (MoA) (Zhang et al., 2022) addresses the instability of applying Mixture-of-Experts (MoE) to attention mechanisms by enforcing shared

key and value projections across all heads, following the MQA design (Vyas et al., 2020). Expert sparsity is applied only to the query transformation. Because all heads use the same key and value mappings, discrepancies between expert-routed queries and the fixed keys/values are avoided: the shared projections are trained to be universally compatible with all query experts. This stabilizes training and enables scaling to many heads. However, MoA inherits a core limitation of MQA: since key and value projections are fixed regardless of head count, scaling the number of heads yields diminishing returns, as only the query pathway benefits from expert specialization.

**Native Sparse Attention**   Native Sparse Attention (NSA) (Yuan et al., 2025) proposes to calculate fine-grained attention only for selected blocks based on proximity scores calculated from aggregated information across tokens from a given block. This reduces the necessity of calculating the entire matrix of attention, and therefore leads to significant efficiency improvements in comparison to the dense baseline. NSA, however, does not sparsify QK transformations, and therefore is not suitable for the design proposed by us in this paper with a lot of small attention heads.

# B   RELATED WORK

The quadratic cost of attention in the 2017 transformer model (Vaswani et al., 2017) has led to a wide body of research on efficient attention variants (Kitaev et al., 2020; Choromanski et al., 2021). Popular alternatives are different linear attention variants that typically use a fixed vector or matrix memory and update it recurrently. The 1992 unnormalised linear Transformers (Schmidhuber, 1992; Katharopoulos et al., 2020; Schlag et al., 2021) trade performance for better computational efficiency. State space models (Gu et al., 2020; 2022; Gu and Dao, 2023) are popular alternatives that offer efficient, parallel training while keeping linear cost and efficient inference. The parallel training requirement forces only a linear recurrent relation between the timesteps. A common characteristic of such models is the relatively small, fixed memory that requires extreme compression. Despite recent progress, these models still underperform quadratic attention on many benchmarks (Arora et al., 2024; Jelassi et al., 2024).

Sparse attention methods aim to mitigate the quadratic cost of full attention by computing attention scores for only a subset of token pairs rather than the full attention matrix. These methods typically employ various heuristics to strategically identify which tokens and token relationships are the most important to process. This is often done by introducing special tokens that serve as higher-level representations of entire chunks of tokens, or by assuming emergent hierarchical structures within the attention patterns. For example, SepLLM (Chen et al., 2024) uses separators in the sentence as special tokens that sparse attention focuses on. Sparse Transformer (Child et al., 2019) uses static attention patterns to reduce computational complexity. Longformer (Beltagy et al., 2020) combines sliding window attention with additionally selected tokens globally available. BigBird (Zaheer et al., 2020) combines sliding window attention and global attention on selected tokens, while additionally including randomly selected tokens in the attention. Streaming LLM (Xiao et al., 2024a) discovers and preserves attention sinks as a necessary component despite their inefficiency and combines them with sliding window attention. Some methods (Liu et al., 2023; Li et al., 2025; Cai et al., 2024; Xiao et al., 2024b; Xu et al., 2025b; Gao et al., 2024; Jiang et al., 2024a) focus on post-training attention reduction, motivated by KV-cache reduction. Hash Attention(Desai et al., 2024) uses top-$k$ selection in the attention scores to induce sparsity and improve efficiency. However, learnable sparse attention that can also be used during training (Yuan et al., 2025) remains important as the quadratic cost of the self-attention mechanism is also problematic in the very costly pretaining phase.

Mixture-of-Experts (MoE) (Shazeer et al., 2017) have emerged as a promising paradigm for scaling model capacity without a proportional increase in computational cost. By adaptively routing input tokens to specialized experts, MoE architectures selectively activate only a part of the network. MoEs applied to transformer feedforward networks (Lepikhin et al., 2021; Fedus et al., 2022) have been widely adapted in LLMs (Guo et al., 2025; Jiang et al., 2024b; Shen et al., 2024b).

A crucial challenge in MoE is to learn a balanced routing, so that experts are utilized uniformly. Imbalanced routing leads to capacity bottlenecks when certain experts become overused while others are completely ignored. This phenomenon is called expert collapse (Shazeer et al., 2017). Most approaches mitigate it by specific losses that penalize polarized expert selection (Lepikhin et al., 2021), while others propose alternative routing methods (Lewis et al., 2021; Roller et al., 2021).

Expert-Choice routing (Zhou et al., 2022) inverts the selection problem, allowing each expert to choose its preferred tokens. This way, Expert-Choice routing achieves perfect load balancing by definition, at the cost that some tokens are ignored and some are overutilized. Expert-Choice routing, however, cannot be directly applied to autoregressive modeling as it uses a non-autoregressive top-$k$ operation over the tokens. MoD (Raposo et al., 2024) proposes methods to transfer nonautoregressive expert choice routing to an autoregressive model. We leave the investigation of their adaptation to MoSA for future work.

MoE is most often applied to the feedforward part of the transformer. In contrast, some works explore MoEs in the attention mechanism to reduce the high computational cost and memory. Mixture-of-Attention Heads(MoA) (Zhang et al., 2022) selects $k$ query transformations for each token and shares a single key and value projections similarly to Multi-Query Attention(MQA) (Shazeer, 2019). MoA allows for increasing the total number of query heads when using MQA without significantly increasing the computational cost. In contrast, MoSA selects tokens that are routed to full heads with separate queries, keys, and values (and consequently, outputs) utilizing perfect load balancing from expert choice routing for efficient sparse attention. This reduces the cost of each attention head significantly more than MoA and does not require MQA (although it might be combined for further benefits, which we leave for future work). Moreover, MoSA allows for KV-cache savings by reducing the number of selected keys, which is not possible with MoA, apart from the MQA benefit of having single KV transformations. SwitchHead (Csordás et al., 2024) reduces the number of heads (and therefore the number of computed attention matrices) by adding internal experts that can compensate for the lower number of heads. This is orthogonal to MoSA and possibly can be combined for further improvements. Multi-head attention as Mixture of Head Attention (Jin et al., 2024) proposes to use dynamic weights for the output projection in order to treat the heads as experts for tokens. However, it requires calculating all attention matrices, lacking the benefits of sparse computation.

Mixture-of-Depths(MoD) (Raposo et al., 2024) selects inputs to pass through a given entire transformer block to allow adaptive computation. This includes the attention mechanism. This produces efficiency gains in an FLOP-limited budget for the entire training. MoSA has multiple selection mechanisms, one for each head, and by increasing the number of heads it processes the sentence in a distributed way - each head processing its own chunk of the sentence.

## C  FLOPS CALCULATION.

Let $T$ be the sequence length, $h$ the hidden dimension of the model, $h'$ the hidden dimension in each head (after passing through the query, key or value projection), $k$ the number of tokens selected for each head, and the sparsity rate $\rho = \frac{T}{k}$.

Multiplying matrices of shape $[i, j]$ and $[j, k]$ takes precisely $(2j - 1)ik$ FLOPs. For simplicity, following common practice, we approximate it by $2jik$.

In the dense attention layer, calculating each projection (e.g., $Q_i = xW_{Q_i}$) requires $2hh'T$ FLOPs. Computing the attention matrix $QK^\top$, and multiplying the attention matrix by values $V$ both cost $2h'T^2$ FLOPs.

Calculating the projections and attention in the MoSA head is identical, except that now we are operating on $k$ tokens instead of $T$. The MoSA head involves an additional routing overhead. Calculating the routing scores costs $2hT$ FLOPs, and multiplying the intermediate values in the matrix $\in \mathbb{R}^{k \times h'}$ by the scores costs an additional $h'k$ FLOPs per head.

FLOPs cost of a single head is equal to:

$$\text{FLOP}_{\text{dense}} = \underbrace{8hh'T}_{\text{Q,K,V,O mappings}} + \underbrace{4h'T^2}_{\text{Attention}}$$

$$\text{FLOP}_{\text{mosa}} = \underbrace{8hh'k}_{\text{Q,K,V,O mappings}} + \underbrace{4h'k^2}_{\text{Attention}} + \underbrace{2hT + h'k}_{\text{routing overhead}}$$

$$\text{FLOP}_{\text{fixed}} = \underbrace{8hh'k}_{\text{Q,K,V,O mappings}} + \underbrace{4h'k^2}_{\text{Attention}}$$

$$\text{FLOP}_{\text{routing}} = \underbrace{6hh'T}_{\text{Q=K,V,O mappings}} + \underbrace{4h'k^2\rho}_{\text{Attention}} + \underbrace{2h'T}_{\text{cluster selection}} = \rho(6hh'k + 4h'k^2) + 2h'T$$

Note that, typically $k << T$, hence the MoSA head is significantly cheaper compared to a dense head.

The selection mechanism in MoSA introduces an additional overhead of $2hT + h'k$ ($2hT$ comes from token scoring and $h'k$ comes from multiplying the output by the scores), which is small compared to the rest. As a consequence, the cost of the MoSA head is comparable to that of the fixed sparsity attention head, while allowing content-based dynamic sparsity.

In contrast to MoSA and fixed attention, the Routing Transformer must compute all tokens by query, key, value, and output transformations. However, in the Routing Transformer for autoregressive text $K = Q$, therefore, only 3 projections need to be computed. Hence, the projection cost is equal to $6hh'T$. The attention in the Routing Transformer has multiple clusters inside each head. More specifically, it has $\rho$ clusters of size $k$, and therefore the attention cost of the head is equal to the attention cost of the cluster multiplied by the number of clusters. The Routing Transformer has an additional layer normalization inside the head, which we omitted for simplicity.

FLOP-wise, one Routing Attention head more or less corresponds to $\rho$ fixed attention or $\rho$ MoSA heads. Loosely speaking, MoSA with $\rho$ heads is similar to the Routing Attention head, where each cluster has its own custom linear transformation, rather than a single one shared among clusters.

For the multihead version, the FLOPs are multiplied by the number of heads $H$. There is an additional cost caused by summing the head contributions to a single output (Equations 2 and 3). However, this is already taken into account by the $2hh'TH$ cost of the output projection for multiple heads: $H(2h' - 1)hT + (H - 1)hT = (2h'H - 1)hT \approx 2hh'TH$.

Note that in the standard notation (Vaswani et al., 2017), the heads are first concatenated and then transformed with a single output projection instead of splitting the output operation into individual head transformations and summing. However, the result and the derivation of the FLOP counts are the same.

In the feedforward block, the intermediate layer has a typical size of $4h$. Therefore, the cost of the block is equal to $16h^2T$. Therefore, the FLOP cost of the forward pass of the entire model with $l$ layers, a hybrid attention with $H_{dense}$ dense heads and $H_{mosa}$ MoSA heads is equal to:

$$lH_{dense}(8hh'T + 4h'T^2) + lH_{mosa}(8hh'k + 4h'k^2 + 2hT + h'k) + 16lh^2T$$

We omit the operations related to layer normalizations, residuals, and token embeddings from the FLOP calculations as they are negligible compared to the rest and represent an identical overhead for both dense and MoSA models. Thus, incorporating them does not influence the FLOP-matching process. This is also true for the feedforward block; yet, we still included it because it constitutes a significant portion of the total cost. We present the FLOP cost of all of our model classes (*Tiny, Small, Medium* and *Large*) in Table 4.

All models are based on the transformer architecture with Pre-layer normalisation(Xiong et al., 2020). Each model class *Tiny, Small, Medium* and *Large* follows the hyperparameters of the dense model. The necessary forward pass FLOPs are calculated according to Sec. C. The number of heads in the sparse models is set so that the resulting model is FLOP-matched to the dense baseline as closely as possible. When this is not perfectly possible, we ensure that its FLOP count never exceeds that of the baseline. For pure MoSA, all heads are replaced with MoSA heads. For the hybrid sparse models, 4 dense heads are kept, and the remaining ones are replaced with sparse heads.

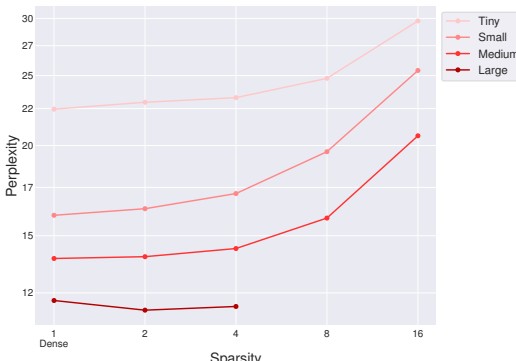

Figure 5: Perplexity of IsoFLOP matching models under pure MoSA setting. Each curve corresponds to a given FLOP budget. For a given sparsity, we replace all dense heads with a FLOP equivalent number of MoSA heads. In contrast to Fig 3, sparse models fail to outperform the baseline (apart from the Large model). This demonstrates the symbiotic relation between dense heads and MoSA heads in the hybrid model.

## D    ANALYSING HYBRID MODELS

While the learned sparse attention can theoretically capture any attention pattern, the introduction of the routing mechanism complicates the learning dynamics. The router and the attention weights must be learned jointly. The router needs to identify relevant token pairs, while the attention weights learn to process these selected interactions. This interdependence can lead to training instabilities, particularly in the early stages, when router decisions are largely random. Poor initial routing can prevent attention heads from learning meaningful patterns, while the lack of meaningful patterns prevents the router from learning to select important tokens, creating a vicious circle.

Our preliminary experiments have shown that pure MoSA models without additional dense heads fail to improve the perplexity of dense baselines. To verify this, we conducted a study similar to our main results in Sec. 3. We gradually increase the sparsity by replacing all dense heads with MoSA heads while maintaining an identical FLOP count to the baseline. We do this by finding the maximum number of MoSA heads for which the FLOP count remains lower than the baseline. The results, shown in Fig. 5, demonstrate that increasing sparsity monotonically worsens model performance in most settings. This performance degradation with pure MoSA heads likely stems from the stability issues explained in the previous paragraph.

Interestingly, the largest model is an exception, and initially there is a visible improvement from 12.20 baseline perplexity to 11.83 perplexity of the FLOP-matched pure MoSA model with sparsity 2. This is still significantly worse than the 11.15 perplexity of the hybrid model with sparsity 4. Moreover, the saturation is much faster than for hybrid models. For hybrid models, the sparsity around 32 or 64 seems to be optimal. In contrast, for the MoSA-only model, the best perplexity is reached for sparsity 2 for the *Large* budget and 1 for the smaller ones. However, the conclusion is consistent across all scales: hybrid MoSA models significantly outperform MoSA-only models, which generally underperform the dense baseline. Thus, hybridization seems necessary.

The impact of sparsification is also visible in the training characteristics. Compared to the baseline, pure MoSA models start to plateau faster. While the losses of dense and hybrid models continue to show steep initial improvement, pure MoSA models slow down much sooner. This supports our hypothesis about the difficulty of learning the routing and attention simultaneously. We compare the training losses in Fig. 6.

**Optimal Number of Dense Heads**    Hybrid models consistently outperform pure MoSA models. This raises a natural question: What is the optimal ratio of dense to sparse heads and how does this ratio relate to the sparsity rate?

To answer these questions, we conducted a series of experiments in which we varied both the sparsity factor of MoSA heads and the number of dense heads while keeping the total FLOP budget constant.

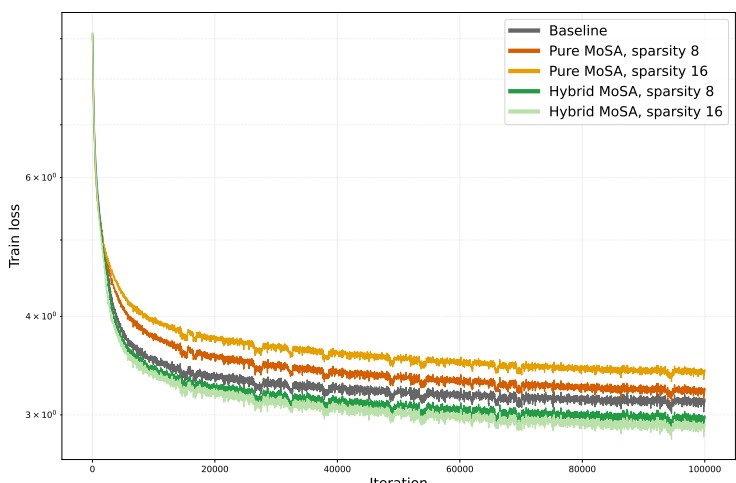

Figure 6: Training losses of the Tiny models comparing the baseline, pure MoSA, and hybrid models. The dense baseline clearly divides the models into two groups: all pure MoSA models perform worse (higher loss), while all hybrid models demonstrate superior performance (lower loss). Notably, increasing sparsity intensifies the difference for both model types: hybrid models achieve progressively lower loss with greater sparsity, whereas pure MoSA models show increasingly higher loss as sparsity increases. Additionally, the early training phase (between 5,000 and 10,000 steps) reveals a distinct pattern where pure MoSA models experience a more rapid slowdown in their learning progress compared to both dense and hybrid models.

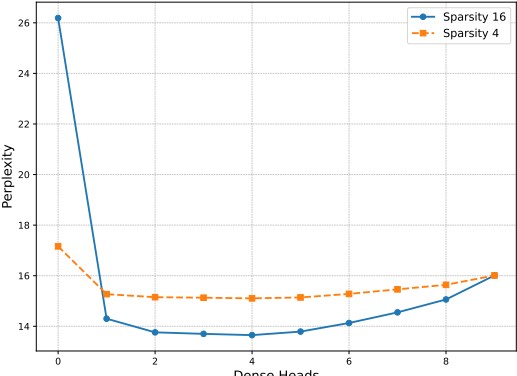

Figure 7: Perplexity of the FLOP matched models with a different number of dense heads for sparsities 4 and 16. 9 dense heads correspond to the dense baseline.

We choose to use the *small* model and investigate sparsities $\rho = 4$ and $\rho = 16$, while we set the number of dense heads in the hybrid model from 0 to 9 (full dense model) and adapt the number of sparse heads to match the FLOP budget. Our results are shown in Fig. 7. We can see that the optimal number of dense heads in this case is 4 and is sparsity-agnostic. Because of this, we chose to use 4 dense heads in our main experiments in Sect. 3. Furthermore, we observe that it is critical to have at least one dense head. Having more than one has diminishing returns, and having more than 4 has a negative effect on the performance. The plot also shows that the lack of dense heads is more hurtful for models with higher sparsities. We conclude that in our case, 4 heads are sufficient to stabilize the training, and it is better to allocate the remaining FLOP budget to the more efficient MoSA heads.

# E  MoSAIC

In addition to expert-routing, we explored a token-choice routing design for MoSA for completeness and demonstration of feasibility. We refer to this variant as Mixture of Sparse Attention with Independent Choice (MoSAIC). Adaptation of MOSA to token-choice routing, which inherently faciliate autoregressive modelling, inevitably introduces two design challenges. Specifically, good balancing of tokens has to be assured by additional losses. We use standard load balancing tokens (Fedus et al., 2021; Lepikhin et al., 2021). In App. E.1 we investigate the impact of load balancing loss on the result and demonstrate the fragility of the model.

Imperfect load balancing has a stronger impact on the token-choice attention experts than on token-choice FFNs, because of the inter-token dependencies in the attention heads. Specifically, output of a token for a given head is influenced by other tokens that are in the head. Because of that, strategies for handling imperfectly balanced loads needs to be carefully considered. We investigate two main decisions in this context. Which tokens to select to the head in case of overloading (and which to pad in case of underutilization), and what to do with the padded tokens (following the argument that they might be harmful for other tokens). We propose and investigate different imperfectly balanced load handling strategies in App. E.2.

## E.1  Load Balancing

In mixture-of-experts architectures with token-level routing, a key challenge is that the routing network may converge to a degenerate solution where only a small subset of experts receives most tokens. This *collapse* not only wastes model capacity but also destabilizes training, as unused experts fail to learn meaningful representations. To counteract this, prior work such as GShard (Lepikhin et al., 2021) and Switch Transformer (Fedus et al., 2021) introduced a load balancing loss term. The purpose of this auxiliary loss is to encourage more uniform expert utilization while still allowing the routing network to specialize experts according to input structure.

**Formulation.**  We adopt the same load balancing loss as in GShard and Switch Transformer. Let $f_i$ denote the fraction of tokens dispatched to expert $i$, and $p_i$ the average routing probability assigned to expert $i$. For a system with $N$ experts, the load balancing loss is defined as:

$$\mathcal{L}_{\text{balance}} = \alpha N \cdot \sum_{i=1}^{N} f_i \cdot p_i. \tag{4}$$

This term is minimized when both the assignment frequencies $f_i$ and the routing probabilities $p_i$ are close to uniform across experts. In practice, the overall training objective augments the language modeling loss with a weighted contribution from $\mathcal{L}_{\text{balance}}$, controlled by the *load balancing loss weight*. This provides a direct mechanism to trade off model perplexity against expert utilization efficiency.

We investigated how the *load balancing loss weight* $\alpha$ influences the perplexity of MoSAIC models across different sizes. The experimental setup follows the configuration described in the preceding sections, with sparsity chosen according to the optimal settings reported in the MoSA paper: 64 experts for the *Tiny* and *Small* models, 32 experts for *Medium*, and 4 experts for *Large*. For each model size, we varied the load balancing weight over $\{0.1, 0.2, 0.4, 0.8\}$ and measured perplexity on the validation set.

The results, summarized in Figure 8, show that performance depends on the choice of load balancing weight, although the sensitivity varies across model sizes. The larger models (*Medium* and *Large*) exhibit only modest changes in perplexity, with differences remaining under one point, but the variations are still noticeable. For the smaller models (*Tiny* and *Small*), the effect is more pronounced: the *Tiny* model in particular benefits from higher load balancing weights, while the *Small* model achieves its best performance around moderate values ($0.4$).

These findings indicate that while the load balancing weight does not dominate performance, it is not entirely negligible either. Its influence is most apparent in the *low-capacity regime*, where balancing expert usage more directly impacts generalization. In larger models, the effect is less dramatic but still measurable, suggesting that careful selection of this parameter can yield incremental gains across all scales.

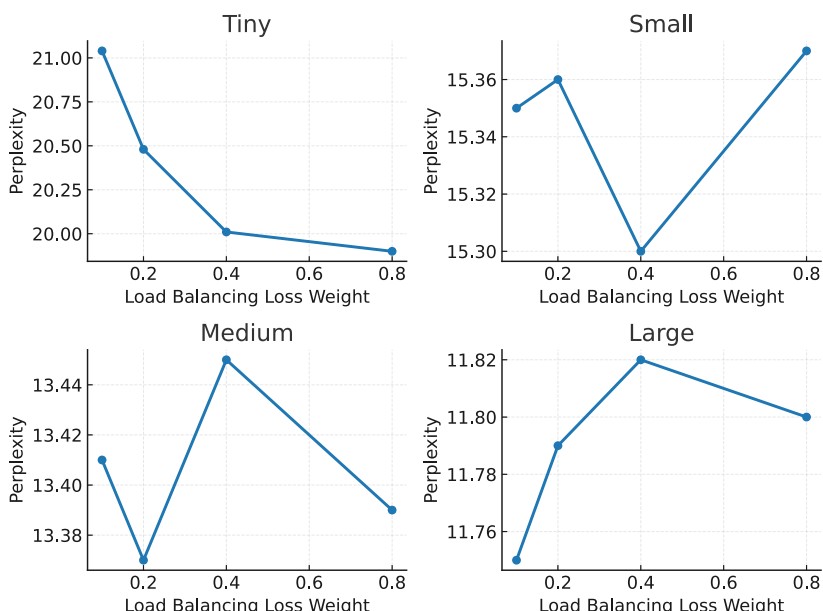

Figure 8: Perplexity of MoSAIC models as a function of the load balancing loss weight $\alpha$. Each subplot corresponds to a different model size (*Tiny*, *Small*, *Medium*, *Large*), with sparsity fixed to the optimal configuration reported in the MoSA paper. Smaller models exhibit significant sensitivity to the choice of $\alpha$. However, larger models show only modest variation.

### E.2 OVERLOADED TOKENS

Unlike mixture-of-experts applied in the feed-forward blocks, our experts are implemented as attention heads. This structural difference has important consequences: assigning a token to an expert not only determines the output representation of that token, but also shapes the interactions with all other tokens routed to the same head. In other words, expert assignment induces an inter-token relation that propagates beyond the individual token's output. As a result, the way overloaded tokens are handled becomes especially critical, since decisions about which tokens to include, ignore, or pad can directly alter the relational structure inside the expert.

To probe this phenomenon, we evaluated the model under different design choices for handling overloaded tokens. Specifically, we compared three strategies: (*include*) retaining all padded tokens and including them in the attention computation (while respecting autoregressive mask), (*identity*) allowing the padded tokens to attend to only to themselves, also make them invisible for other tokens. Effectively it transforms the padded tokens directly through V and O transformations. Finally, (*ignore*) - masking out the padded tokens entirely. Each of these options reflects a different trade-off. Inclusion allows the most efficient use of the computation, as the padded tokens will take space in the batch anyway. The identity still allows to use the computation for the padded tokens, without risking that due to their appearance in the batch without being selected they will disrupt the integrity of the expert transformation. Ignore is the safest, clearest theoretical option that transforms only tokens that actually selected given expert.

The results for both the *Tiny* and *Medium* models (Figure 9) demonstrate that these choices yield measurable differences in perplexity. For retaining tokens via inclusion tends to be the best option, while ignoring tokens degrades performance more noticeably. Furthermore, keeping tokens based on their position (prioritizing early tokens) is most often beneficial. We hypothesize that this might be related to the attention sinks that often appear at the beginning of the sequence.

Taken together, these findings emphasize that the semantics of expert assignment in attention-based MoEs make overload handling a first-class design decision. Unlike in feed-forward MoEs where dropping a token primarily affects its own representation, here the effect propagates to all tokens in the expert, making the system more sensitive to overload strategies. This suggests that future work

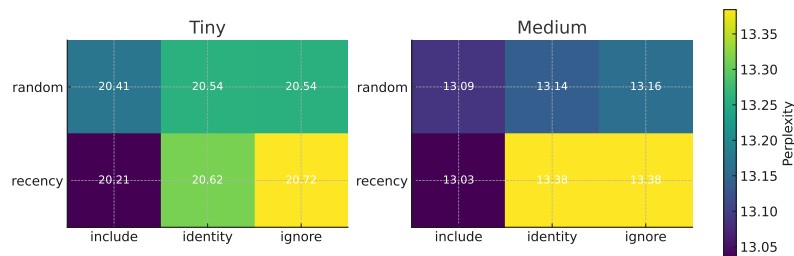

Figure 9: Perplexity of *Tiny* and *Medium* MoSAIC models under different strategies for handling overloaded tokens. The two axes correspond to the choice of routing priority (*random* vs. *recency*) and padded operation (*include*, *identity*, *ignore*). Because experts are attention heads, these design choices affect not only the overloaded tokens themselves but also the inter-token relations within each expert. The results show that the optimal strategy varies across model sizes, highlighting the importance of tuning overload handling rather than treating it as a fixed implementation detail.

on scaling attention-based mixtures must account for overload handling not merely as an engineering detail, but as a factor shaping the model's representational behavior and generalization.

## F    DOWNSTREAM TASKS

We evaluate the zero-shot downstream performance of MoSA on six established benchmarks: LAM-BADA (Paperno et al., 2016), WinoGrande (Sakaguchi et al., 2020), BLiMP (Warstadt et al., 2020), HellaSwag (Zellers et al., 2019), PIQA (Bisk et al., 2020) and AI2ARC (Clark et al., 2018)—covering tasks from cloze-style completion to commonsense reasoning.

During training, MoSA operates on sequences of more or less constant size $T = 1024$. However, for downstream tasks, some inputs will be much shorter. For example, most datapoints in the BLiMP dataset do not exceed 10 tokens. In order to handle such situations, we adaptively choose the number of tokens for each input to be $k = \max(\lfloor \frac{T}{\rho} \rfloor, 2)$ tokens for each head. This simulates the ratio of tokens selected for the attention head during the training. Moreover, it ensures that at least 2 tokens are selected, which is the minimum necessary for the attention to model any cross-token dependencies.

For each scale and sparse model type, we select the model with sparsity $\rho > 1$ that produced the best perplexity in the IsoFLOP scenario (Sec. 3). We also include the dense baseline for each size. Table 3 reports the performance across the tasks. The best result for a given task across model types is bold.

For *Tiny*, *Small*, and *Medium* scales, MoSA generally outperforms other models. BLiMP stands as a notable exception, where MoSA consistently underperforms. This weak performance on BLiMP can be attributed to the extremely short length of most examples in the dataset. With longer sequences seen during training, each MoSA head can selectively process only the tokens it handles well. However, in short sequences, the shortage of tokens forces MoSA heads to operate on tokens outside their training distribution. Furthermore, when $\lfloor \frac{T}{\rho} \rfloor = 1$, resulting in only 2 tokens being selected, there is a significant discrepancy between the percentage of selected tokens compared to training conditions. Models with a high sparsity factor of $64$ typically select only $1.56\%$ tokens in a sequence for each attention head. Yet for a sequence length of $T = 10$, 2 selected tokens represent $20\%$ of the sentence, creating a distribution mismatch.

Moreover, in *Large* scale, the Dense baseline outperforms MoSA despite having much higher perplexity. We attribute the downstream performance gap of MoSA to two main factors. First, MoE architectures have been shown to suffer from expert overspecialization, which often leads to decreased performance in downstream tasks (Fedus et al., 2022; Zoph et al., 2022). Instruction tuning has been shown to mitigate this issue (Shen et al., 2024a).

Furthermore, content-based sparse attention methods tend to struggle on shorter sequence[‡]. Our experiments confirm this pattern, as MoSA outperforms the Routing Attention in most tasks. Furthermore, some runs of the Routing Attention were unstable in context of downstream tasks (Medium scale of the Routing Attention). Practitioners report that extending training by additional epochs on truncated sequences can mitigate the issues of sparse attention methods on short sequences[‡].

|        | Model   | LAMBADA | WinoGrande | BLiMP | HellaSwag | PIQA | AI2ARC |
|--------|---------|---------|------------|-------|-----------|------|--------|
| Tiny   | Dense   | 18.7    | 50.3       | 72.0  | 27.5      | 59.4 | 28.0   |
|        | Routing | 14.0    | 51.3       | 66.2  | 27.8      | 57.1 | 25.9   |
|        | Fixed   | 17.1    | 50.6       | **72.5** | 27.7   | 58.6 | 28.1   |
|        | MoSA    | **26.5** | **53.0**  | 65.5  | **29.1**  | **59.7** | **29.4** |
|        | MoSAIC  | 22.7    | 52.2       | **72.5** | 28.2   | 58.0 | 27.9   |
| Small  | Dense   | 25.8    | 52.1       | 76.2  | 30.9      | 62.4 | 30.1   |
|        | Routing | 19.2    | 50.7       | 70.2  | 28.0      | 57.6 | 27.3   |
|        | Fixed   | 24.6    | 51.6       | 75.3  | 30.1      | **63.2** | 30.2 |
|        | MoSA    | **29.4** | 51.9      | 70.5  | **31.9**  | **63.2** | 30.0 |
|        | MoSAIC  | 26.8    | **52.2**   | **77.6** | 31.0   | 62.6 | **30.3** |
| Medium | Dense   | 31.4    | 51.2       | 77.8  | 33.8      | 64.5 | **31.5** |
|        | Routing | 10.2    | 51.5       | 65.9  | 30.3      | 57.8 | 27.8   |
|        | Fixed   | 29.4    | 51.4       | 77.3  | 33.0      | **64.6** | **31.5** |
|        | MoSA    | **36.1** | **52.1**  | 66.1  | **34.2**  | 63.3 | 31.4   |
|        | MoSAIC  | 31.8    | **52.1**   | **78.7** | 33.9   | **64.6** | 31.4 |
| Large  | Dense   | 36.2    | 52.5       | **80.4** | **38.7** | **67.1** | **33.8** |
|        | Routing | 27.5    | 51.1       | 76.5  | 36.2      | 64.1 | 32.5   |
|        | Fixed   | 32.3    | 51.7       | 79.6  | 35.9      | 66.0 | 32.2   |
|        | MoSA    | 35.0    | 51.4       | 74.2  | 37.5      | 65.9 | 31.7   |
|        | MoSAIC  | **40.4** | **52.7**  | 79.5  | 37.4      | 65.4 | 32.6   |

Table 3: Accuracy on downstream zero-shot tasks. Each model is selected with the best sparsity in the IsoFLOP comparison. Note that on downstream tasks, the token selection mechanism of MoSA operates out of distribution. Despite this, MoSA often outperforms the dense baseline. Even when it does not, the performance gap is usually small. MoSAIC partially mitigates these issues by providing a token-choice routing mechanism suitable for autoregressive, although residual gaps suggest that some limitations are inherent to content-based sparse attention.

## G  LLM USAGE

Large language models (LLMs) were used only to help edit the text. They improved grammar, style, and readability, but did not play any role in shaping ideas, designing experiments, analyzing results, or creating new content. All scientific work, including the concepts, methods, and interpretations, was done entirely by the authors.

## H  DETAILS OF THE MODELS

In the Table 4 we list hyperparameters all of dense baselines.

**Implementation details**  We use the SentencePiece (Kudo and Richardson, 2018) tokenizer based on sub-word units (Sennrich et al., 2016; Schuster and Nakajima, 2012) a vocabulary size of 8000. All

---

[‡]See: `https://github.com/lucidrains/routing-transformer?tab=readme-ov-file#issues`

our models are trained on the C4 (Raffel et al., 2020) dataset for 100k batches, with batch size $B = 64$ and sequence length $T = 1024$. This means that we train on the $10^5 SB \approx 6.5B$ tokens from the dataset. We use the Adam (Kingma and Ba, 2015) optimizer with a learning rate of 0.00025, gradient clipping above the norm of 0.25, and a linear warmup for 4k steps. For detailed hyperparameters, please refer to Appendix H.

**Positional encodings.** All our experiments use Rotary Positional Encodings (RoPE) (Su et al., 2021). RoPE applies positional encodings for each attention head after query and key mapping. It does this by rotating them at an angle determined by the token's position in a sentence. Similarly to the attention mask, we must ensure that the rotations correspond to the token's original position in the sequence $X$ rather than the selected subset $\boldsymbol{X}^S$. Thus, we adapt RoPE to be aware of token positions $I$. Following standard practice, we rotate half of the dimensions and leave the other half unchanged.

**Resources Used.** All the experiments in the paper were run on NVIDIA-A100 80GB nodes with GPUs ranging from 1 for small experiments to 4 GPUs. Each experiment was limited by 24h operation time.

|  | Tiny | Small | Medium | Large |
|---|---|---|---|---|
| FLOPs per pass (G) | 54.76 | 219.85 | 430.70 | 1,130.65 |
| Layers | 6 | 9 | 18 | 27 |
| Hidden size | 512 | 1,024 | 1,024 | 1,280 |
| Feedforward hidden size | 2,048 | 4,096 | 4,096 | 5,120 |
| Head hidden size | 64 | 64 | 64 | 64 |
| Number of heads | 9 | 9 | 9 | 16 |

Table 4: Hyperparameters of the different model variants and the corresponding FLOP cost of the forward pass for a sequence length of $T = 1024$.

| | | Sparsity | | | | | | | | |
|---|---|---|---|---|---|---|---|---|---|---|
| | | 1 | 2 | 4 | 8 | 16 | 32 | 64 | 128 | 256 |
| | | Perplexity ($\downarrow$) for given sparsity | | | | | | | | |
| Tiny | MoSA | 22.46 | 21.76 | 20.45 | 19.24 | 18.00 | 16.90 | **16.37** | 17.27 | 18.06 |
| | Pure MoSA | **22.46** | 22.96 | 23.30 | 24.78 | 29.76 | - | - | - | - |
| Small | MoSA | 16.01 | 15.69 | 15.10 | 14.33 | 13.68 | **12.97** | 13.30 | - | - |
| | Pure MoSA | **16.01** | 16.35 | 17.16 | 19.61 | 25.41 | - | - | - | - |
| Med. | MoSA | 13.95 | 13.52 | 12.81 | 12.16 | 11.47 | **11.22** | - | - | - |
| | Pure MoSA | **13.95** | 14.03 | 14.40 | 15.87 | 20.63 | - | - | - | - |
| Large | MoSA | 12.20 | 11.67 | **11.15** | - | - | - | - | - | - |
| | Pure MoSA | 12.20 | **11.83** | 11.97 | - | - | - | - | - | - |

| | | Number of parameters for given sparsity | | | | | | | | |
|---|---|---|---|---|---|---|---|---|---|---|
| Tiny | MoSA | 28M | 34M | 48M | 78M | 136M | 242M | 423M | 693M | 1B |
| | Pure MoSA | 28M | 39M | 65M | 119M | 222M | - | - | - | - |
| Small | MoSA | 113M | 127M | 163M | 229M | 360M | 599M | 1B | - | - |
| | Pure MoSA | 113M | 142M | 203M | 324M | 559M | - | - | - | - |
| Med. | MoSA | 210M | 239M | 310M | 442M | 703M | 1.2B | - | - | - |
| | Pure MoSA | 210M | 267M | 390M | 632M | 1.1B | - | - | - | - |
| Large | MoSA | 516M | 650M | 943M | - | - | - | - | - | - |
| | Pure MoSA | 516M | 703M | 1B | - | - | - | - | - | - |

| | | Number of MoSA heads for given sparsity | | | | | | | | |
|---|---|---|---|---|---|---|---|---|---|---|
| Tiny | MoSA | 0 | 13 | 31 | 69 | 142 | 276 | 505 | 848 | 1277 |
| | Pure MoSA | 0 | 23 | 56 | 124 | 255 | - | - | - | - |
| Small | MoSA | 0 | 11 | 26 | 54 | 109 | 210 | 381 | - | - |
| | Pure MoSA | 0 | 21 | 47 | 98 | 197 | - | - | - | - |
| Med. | MoSA | 0 | 11 | 26 | 54 | 109 | 210 | - | - | - |
| | Pure MoSA | 0 | 21 | 47 | 98 | 197 | - | - | - | - |
| Large | MoSA | 0 | 27 | 60 | - | - | - | - | - | - |
| | Pure MoSA | 0 | 37 | 80 | - | - | - | - | - | - |

Table 5: Detailed statistics of the main IsoFLOP experiments from Sec. 3. Models Tiny, Small, Medium, and Large are as described in App.H. Sparsity 1 corresponds to dense baselines. Pure MoSA models for sparsities $\geq 1$ have only MoSA heads, calculated as the biggest number of heads that will not increase the FLOP budget of the dense baseline (other hyperparameters stay the same as in the baseline). MoSA models have 4 dense heads and the rest of the heads are sparse, calculated such that the flop cost of both dense and sparse heads is lower than the baseline. Therefore, the total number of heads in hybrid models (with sparsity $\geq 1$) is the number shown in the bottom table + 4. For perplexity, the best result for each row is bold.

This passage beautifully captures the concept of mindfulness and the art of being present in the moment. It highlights the peacefulness and liberation found in the quiet spaces between thoughts, where one can embrace the emptiness without the demands of productivity or purpose. This moment of "pure nothingness" serves as a reminder of the importance of taking a step back from the constant stream of thoughts and activities that often fill our lives. By finding solace in the absence of activity, we allow ourselves to experience a different kind of richness—a connection to the simplicity of just being. This can be a powerful form of meditation, helping to rejuvenate the mind and spirit, and offering a respite from the relentless pursuit of meaning and achievement.

