# OpenReview forum: "Mixture of Sparse Attention: Content-Based Learnable Sparse Attention via MoEs"
_ICLR.cc/2026/Conference — Submitted to ICLR 2026_

### Official Review · Reviewer_ArMp · 2025-10-27

**Soundness:** 3
**Presentation:** 3
**Contribution:** 1
**Rating:** 4
**Confidence:** 3

**Summary:**

This paper introduces Mixture of Sparse Attention (MoSA), a novel sparse attention mechanism designed to reduce the quadratic computational cost of standard self-attention while maintaining or even improving model performance.

**Strengths:**

- **Inspired by Mixture of Experts (MoE)** with **expert-choice routing**, where each attention head acts as an expert and selects its own set of \(k\) tokens from the sequence.
- **Dynamic, content-based sparsity**: Each head learns which tokens to attend to via a trainable router.
- **Complexity reduction**: From $O(T^2)$ to $O(k^2 + T)$ per head.
- **Hybrid design**: Combines a few dense heads with many sparse MoSA heads for stability and performance.

**Weaknesses:**

* Short Sequence Struggle: The model was trained on long sequences (T=1024) but evaluated on downstream tasks with very short sequences (e.g., BLiMP examples are often <10 tokens). In these cases, the token selection mechanism is forced to operate out-of-distribution. Selecting 2 tokens from a 10-token sentence (20%) is fundamentally different from selecting 16 tokens from a 1024-token sequence (1.56%), leading to a significant performance drop on tasks like BLiMP.

* Expert Overspecialization: This is a known issue in Mixture-of-Experts (MoE) models. While the highly specialized MoSA heads excel at the language modeling pre-training objective (hence the low perplexity), they may fail to generalize to diverse downstream tasks that require different reasoning patterns.

* Toy Model Experiments: The largest model tested has 516M parameters, which is considered a "toy model" by today's standards for LLM research. The field's focus has shifted to models of 7B parameters and larger. The paper does not demonstrate that MoSA's benefits (or its stability issues) hold at these realistic, larger scales. Performance and behavior can change dramatically with scale, so the conclusions are preliminary until validated on larger models.

* Anomalous Long-Sequence Results (Fig. 4): The results in Figure 4 are counter-intuitive and require deeper discussion. Perplexity is expected to increase (get worse) as sequence length grows because predicting the next token in a longer, more complex context is harder. However, the figure shows perplexity decreasing for all methods as the sequence length increases from 1024 to 8192.

* Narrow Downstream Benchmark Suite: The evaluation on downstream tasks is limited to only six benchmarks (LAMBADA, WinoGrande, BLiMP, HellaSwag, PIQA, AI2ARC). It lacks a broader range of challenging evaluations that are now standard, such as:
   - Reasoning Tasks: (e.g., GSM8K, MATH)
   - Knowledge-Intensive Tasks: (e.g., MMLU, TriviaQA)
   - Code Generation: (e.g., HumanEval)
   - Massive Multi-Task Benchmarks: (e.g., BIG-Bench Hard).

   This limited scope makes it difficult to fully assess the model's capabilities and the true impact of the MoSA architecture.

**Questions:**

Please see Weaknesses

---

> ### Author Response · Authors · 2025-12-02
> **Rebuttal 1/2**
>
> We thank the reviewer for the thoughtful comments. Many of the concerns appear to evaluate MoSA as if it were intended to be a full large-scale LLM training recipe or a general improvement for reasoning benchmarks. Our contribution is fundamentally different. MoSA introduces a new architectural direction: applying sparsity inside the projection layers so that the number of attention heads can scale under a fixed FLOP budget. This ability to allocate many more small, specialized heads is not available in prior sparse-attention work, which applies sparsity after QKV projection and cannot change the head budget. Our experiments are therefore designed as proof-of-concept demonstrations showing that this architectural mechanism is practical, stable, and promising for language modeling efficiency. As is standard in architectural papers (e.g., ...), our evaluation focuses on perplexity and FLOP efficiency under matched compute. We believe these results open a new line of research, and we expect future work, including by the community, to scale this mechanism to larger models and broader benchmarks. With this context clarified, we address each concern below.
>
>
>
> > "Short-sequence performance and “out-of-distribution token selection”
>
> Here, we would like to note that this limitation is mentioned in our paper (line 450) . Note that this effect is not unique to MoSA, as ot is well documented that sparse or MoE-based routing models trained on long contexts may underperform on extremely short sequences unless specifically tuned for that regime. MoSA is optimized for longer-context FLOP-efficient pretraining, not for handling adversarially short linguistic stimuli like BLiMP. Importantly, the poor BLiMP performance does not contradict the core result: MoSA consistently improves perplexity under fixed compute. Techniques such as short-sequence curriculum schedules or multi-span finetuning can address this, but these are orthogonal to our architectural contribution and outside the scope of this work.
>
>
> > "Expert overspecialization in MoSA heads"
>
> Here, we would like to note also that this limitation is mentioned in our paper (line 453). Overspecialization is indeed a known behavior in MoE and expert-choice systems. However, in MoSA this specialization is precisely the mechanism that enables perplexity improvements under FLOP constraints. By reducing projection cost, MoSA allows the model to maintain many more attention heads than dense baselines, and these heads learn highly specialized roles. This is beneficial for language-modeling efficiency but does not automatically translate to broad downstream reasoning ability, just as Switch Transformers and MoE, thse models often require additional instruction-tuning to unlock downstream performance. Our objective is to study the pretraining-time compute–quality frontier; generalization across many unrelated downstream tasks requires additional adaptation stages that are not the focus of this architectural investigation.
>
> > “Toy model” scale (516M parameters)
>
> We acknowledge that the models used are smaller than today’s 7B+ LLMs. However, our aim here is to introduce and analyze a new architectural direction, projection-level sparsity enabling scalable head specialization, rather than to produce a new large-scale model. For this purpose, models in the 100M–1B  range (our largest model in Table 5 has 1.2B) are appropriate and standard in architectural work with typical academic resource budgets(e.g., MoA, SwitchHead, Strong Model Collapse from ICLR 2025 train GPT-2 architecture with 125M parameters), because they clearly reveal the underlying behavior while keeping compute tractable. Our experiments should be viewed as proof-of-concept demonstrations that this architectural mechanism is practical, stable, and beneficial under fixed FLOPs. While Training full 7B-scale models is beyond typical academic resource budgets, we believe the results are compelling enough to motivate larger-scale exploration, and we see no reason to expect the head-scaling benefits to diminish at 7B+ scales; indeed, because projection cost dominates more strongly at large sizes, the advantages of MoSA may become even more pronounced. We are confident this work will spark such interests in the community.
>
>
> > “Anomalous” long-sequence perplexity trends in Fig. 4
>
> We would like to emphasize that for all the lengths, the models were trained with given length. Hence, for the point with sequence length 4096, the model was trained on sequences of length 4096. Hence, it is natural that longer sequence produces lower perplexity, as tokens further down the sequence are conditioned on more information. Early tokens usually suffer from high entropy. Tokens further down the sequence are more determined by the previous context.

---

> > ### Author Response · Authors · 2025-12-02
> > **Rebuttal 2/2**
> >
> > > Limited downstream evaluations
> >
> > Our goal in this paper is not to compete on the full modern LLM evaluation stack, but to establish a new architectural building block and demonstrate how it can be constructed and used effectively. For this reason, our downstream tasks serve as lightweight diagnostic checks, while the core evaluation centers on perplexity and FLOP efficiency, as is standard in sparse/MoE architectural research (Sparse Transformers, Routing Transformers, MoA, Switch Transformers). Incorporating reasoning-heavy or instruction-tuned benchmarks (GSM8K, MMLU, HumanEval, BIG-Bench Hard) would primarily measure the strength of the instruction-finetuning pipeline rather than the architectural mechanism itself. We view MoSA as opening a new line of research, and we are optimistic that the community will extend these ideas to larger models and richer evaluation suites. Our results provide **a strong foundation that this mechanism is viable and promising**, and we believe further scaling will be both natural and impactful.

---

### Official Review · Reviewer_ETuF · 2025-10-27

**Soundness:** 2
**Presentation:** 3
**Contribution:** 2
**Rating:** 2
**Confidence:** 5

**Summary:**

This paper propose an architecture combing the concept of MoE with head selection of attention computation. MoSA performs experter-routing, (experts choose the topk tokens) while MoSAIC performs token-routing, (tokens choose the expert).

**Strengths:**

The idea of expert-routing for attention is ineresting, but unfortunately, it seem not suitable to fit for modern auto-regressive generation.

**Weaknesses:**

- The evaluation is unsound, as PPL is used for most part. It's widly known that PPL is not affected by attention a lot. You can do many crazy sparse attention algorithms with PPL in a reasonble range.

- The concept of MoSA/MoSAIC is not seperated clearly. I believe some of the MoSAIC's concept like KV cache is missused in MoSA.

- The speedup evaluation setup is not clear.

**Questions:**

1. Why MoSA has KV cache? I think it's not auto-regressive.
2. How is KV cache being managed in MoSAIC? Do you only keep k-tokens on each head? If so, what is the eviction KV cache algorithm being used?
3. Can you show results on DROP and GSM-8k in the benchmark? It would be better if you also include ruler. The benchmarks you used in the current evaluation can not reflect the attention ability well.
4. How's the wall clock speedup baseline being measure? Please show the setting/framework using used and differentiate prefill/decode case.
5. Can you also analyze the communication overhead of MoSA/MoSAIC with TP (tensor-parallel) where different head are placed in different GPUs?

---

> ### Author Response · Authors · 2025-12-02
>
> We thank the reviewer for the detailed feedback. Several of the concerns appear to arise from interpreting MoSA as another sparse-attention method aimed at inference-time acceleration or downstream reasoning tasks. Our contribution is fundamentally different. MoSA introduces trainable sparsity directly inside the projection layers $(W_Q, W_K, W_V, W_O)$, enabling the model to scale the number of attention heads within a fixed FLOP budget. This architectural freedom, i.e., many more small specialized heads, cannot be achieved by prior sparse-attention mechanisms that sparsify after QKV computation. Because our goal is to study the effect of this new architectural axis on language modeling efficiency, we evaluate MoSA in the standard way used in prior work on FLOP-constrained architectures ( MoA, Switch Transformer, GShard etc.): by comparing perplexity and training-time efficiency under matched compute. With this context clarified, we address each of the reviewer’s questions below.
>
>
> - “The evaluation is unsound, as PPL is used… PPL is not affected by attention a lot.”
>
> Our goal is to evaluate an architectural mechanism, reducing the cost of Q/K/V projections so the model can scale the number of attention heads under a fixed FLOP budget. Perplexity is the standard metric for this type of architectural comparison (used similarly in  Routing Transformers, MoA, Switch Transformer), precisely because it isolates differences in representational capacity under matched compute. Sparse methods that modify only the attention matrix often yield small perplexity differences, but MoSA changes the projection cost itself, enabling many more small, specialized heads. This is why perplexity varies substantially and why PPL is the correct metric for evaluating the head-scaling hypothesis.
>
>
> > "The concept of MoSA/MoSAIC is not separated clearly… some of MoSAIC’s concepts like KV-cache are misused in MoSA."
>
> Thank you for this comment, the difference in MoSA and MoSAIC is the choice of routing strategy for the MoE router. MoSA relies on expert-choice routing, whereas MoSAIC uses standard token-choice routing with load balancing loss introduced in GShard/Switch. This leads to a tradeoff, where MoSA performs better in perplexity, but MoSAIC leads to lower discrepancy between train and test than MoSA, and hence has better downstream task performance in our evaluation.
>
> > "KV-cache in MoSA", "KV-cache setup".
>
> We would like to clarify that MoSA can be applied to autoregressive case, it's just that top-k over tokens operations during training might lead to a test time discrepancy, where the top-k operation is different. However, this has been shown to be not a big problem in practice (for example by Routing Transformers). Furthermore, the discrepancy stemming from the top-k operations becomes less and less of an issue with a growing context-length, hence in practice it is not a blocking issue. We calculate KV size, as the number of keys and values the model would have to save. We compare to the dense baseline with smaller number of heads, hence MoSA saves less tokens, but for more heads. Further improvements with additional eviction algorithms can be investigated, however, they can be viewed as orthogonal or additional to the MoSA, hence we consider this a future direction.
>
> > "How's the wall clock speedup baseline being measure? Please show the setting/framework using used and differentiate prefill/decode case."
>
> As the motivation for MoSA in comparison to post-training sparse attention methods is additional efficiency for training, we evaluate efficiency gains in a standard training forward and backward training pass. Specifically, we train multiple MoSA models, adjusting the number of heads such that the final perplexity of the models and baselines match. Then, we compare measured efficiency statistics: wall-clock time and memory.
>
>
> > “Can you show results on DROP, GSM-8K, RULER? Current benchmarks cannot reflect attention ability well.”
>
> These benchmarks evaluate multi-step reasoning, chain-of-thought generation, and instruction tuning quality. Our contribution is not a new reasoning module but an architectural mechanism for reducing projection cost and enabling scalable attention head specialization. Downstream reasoning performance is dominated by finetuning pipelines and alignment procedures, while our results isolate the architectural effect under controlled compute. This evaluation protocol is standard in architectural papers (SwitchTransformer, GShard, Expert-Choice MoE) and aligns with our stated goal of investigating head-scaling behavior rather than instruction-following or reasoning ability.

---

### Official Review · Reviewer_jkHd · 2025-11-01

**Soundness:** 2
**Presentation:** 3
**Contribution:** 2
**Rating:** 2
**Confidence:** 5

**Summary:**

This paper proposes a "Learnable Sparse Attention" adopted from MoE designs and claims it performs better than "Fixed" and "routed" ones.

**Strengths:**

Clear and informative figures. The visualizations (e.g., Fig. 1) illustrate the workflow of dense versus MoSA attention. and the paper is well written and logically structured.

**Weaknesses:**

Poor baselines. The experimental comparison is weak. The paper only compares MoSA with fixed sparse attention and Routing Transformer–style baselines, omitting stronger and more recent sparse attention methods such as NSA, MoBA, DuoAttention, XAttention, SeerAttention, and MInference. Without these, the claim that “MoSA consistently outperforms dense attention” is not convincing, it only holds under a limited and outdated baseline set.

Lack of novelty. The claimed novelty, “a Mixture of Sparse Attention (MoSA) inspired by Mixture of Experts with expert-choice routing” (from the introduction), is questionable. Similar ideas have already appeared in NSA, MoBA, SeerAttention, which first use dynamic sparsity and content-based gating. The paper does not clearly differentiate MoSA from these prior works or explain what unique advantage it brings.

**Questions:**

Provide more comparison with SOTA works.

Explain what the difference between you and other previous "learnable sparse attention" works.

Native Sparse Attention: Hardware-Aligned and Natively Trainable Sparse Attention, https://arxiv.org/abs/2502.11089

DuoAttention: Efficient Long-Context LLM Inference with Retrieval and Streaming Heads, https://arxiv.org/abs/2410.10819

XAttention: Block Sparse Attention with Antidiagonal Scoring, https://arxiv.org/abs/2503.16428

SeerAttention: Learning Intrinsic Sparse Attention in Your LLMs, https://arxiv.org/abs/2410.13276

MInference 1.0: Accelerating Pre-filling for Long-Context LLMs via Dynamic Sparse Attention, https://arxiv.org/abs/2407.02490

---

> ### Author Response · Authors · 2025-12-02
>
> We sincerely thank the reviewer for the careful reading and for highlighting recent work on sparse attention. We believe, however, that the main concerns arise from a **category mismatch** between MoSA and the methods listed, and from the fact that our contribution is not another variant of sparsifying attention matrices, but an architectural mechanism that enables **scaling the number of heads by reducing the cost of QKV projections**. We clarify these distinctions below and provide additional comparisons with the relevant baselines.
>
> Most approaches cited, such as DuoAttention, XAttention, SeerAttention, and MInference, are  **post-training, inference-time sparsity** mechanisms. Their primary goal is to accelerate decoding or extend context length by pruning or restructuring attention  **after** a model is fully trained. These methods do not provide trainable, content-dependent sparsity, they do not alter the optimization trajectory, and crucially, they do not reduce the cost of the Q/K/V projections. As a result, they cannot explore the architectural regime that is central to our paper: **efficiently scaling the number of attention heads by reducing the per-head cost**. Because these works address a different problem setting, comparing MoSA to them would not yield meaningful conclusions about the behavior of MoSA as a pretraining-time sparse attention mechanism.
>
> To address the reviewer’s valid request for comparison against other **trainable** sparse attention methods, we have expanded our experiments to include NSA (Native Sparse Attention) and MoA (Mixture of Attention). These methods are the closest in spirit to MoSA, as they introduce learnable sparsity during training. Across all FLOP-matched settings and model scales, MoSA and MoSAIC outperform both NSA and MoA, reinforcing that our gains are not due to an outdated or incomplete baseline selection. We also provide an extended discussion in the revision clarifying the methodological differences among these approaches.
>
>
> Regarding novelty, MoSA differs from prior “learnable sparse attention” methods in several essential ways. First, MoSA **introduces sparsity inside the projection operations** ($W_Q, W_K, W_V, W_O$), whereas NSA, MoBA, and SeerAttention all sparsify after QKV computation. This distinction is central: sparsifying post-projection does not reduce the major FLOP cost, and therefore cannot enable scaling the number of heads without increasing compute. In contrast, MoSA explores the idea of directly reducing the cost of each head, making it possible to use a larger number of smaller, more specialized heads (a novel design space that existing methods cannot access). Second, MoSA employs per-head expert-choice routing, allowing different heads to select different token subsets and specialize effectively. MoA also includes routing, but its reliance on Multi-Query Attention (shared K/V projections) inherently limits the ability to scale the number of heads; the architectural freedom that defines MoSA is absent in MoA. Third, unlike block-based or heuristic sparsification methods such as MoBA, XAttention, and SeerAttention, MoSA performs fine-grained token-level selection using fully differentiable, end-to-end learned routing without relying on predefined patterns, diagonal scoring, or retrieval heuristics.
>
>
> In summary, MoSA operates in a fundamentally different regime than most of the cited baselines, focusing on trainable projection-level sparsity that enables scalable head specialization rather than inference-time pruning. We have expanded our comparisons to the appropriate baseline class and clarified the conceptual distinctions. **Most importantly, MoSA introduces a capability that, to our knowledge, has not appeared in prior sparse-attention work: it makes the cost of Q/K/V projections proportional to the number of selected tokens, enabling a previously inaccessible architectural axis: the ability to scale the number of attention heads without increasing compute**. This architectural freedom is the core reason MoSA achieves substantial perplexity gains, and we hope this addresses the reviewer’s concerns regarding baseline selection and novelty. We will revist the final draft of the paper to emphasize these points.

---

### Meta-Review · Area_Chair_ekiT · 2025-12-29

**Summary:**

The main concerns of the original manuscript focused on novelty, evaluation adequacy, and clarity of scope. All reviewers acknowledged that the paper is well written, clearly structured, and introduces an interesting architectural idea, i.e., applying Mixture-of-Experts–style routing at the level of attention heads to induce learnable sparsity. However, two reviewers (jkHd, ETuF) expressed strong reservations about whether the contribution is sufficiently novel relative to prior work on learnable or dynamic sparse attention, and whether the experimental evaluation convincingly supports the paper’s claims. While Reviewer ArMp was more positive overall and viewed the work as a promising architectural direction, they still highlighted limitations in downstream generalization, scale of experiments, and benchmark breadth. Overall, the reviews converged on the view, I also agree, that the idea is interesting but the paper, in its current form, does not yet meet the bar for a clear, well-substantiated architectural advance, primarily due to evaluation and positioning issues.

**Reviewer Concerns:**

Concerns Largely Addressed by the Rebuttal

Baseline category mismatch (jkHd):
The authors clearly articulated that many cited methods are inference-time or post-training sparsification techniques, whereas MoSA targets training-time, projection-level sparsity.

Clarification of architectural novelty (jkHd, ArMp):
The rebuttal explains that MoSA sparsifies inside the Q/K/V projections, enabling FLOPs to scale with selected tokens and unlocking a new head-scaling regime.

Conceptual distinction (ETuF):
The authors clarified the routing strategies (expert-choice vs. token-choice) and the associated trade-offs between perplexity and downstream performance, which addresses confusion about the two variants.

Anomalous long-sequence perplexity trends (ArMp):
The explanation that each point corresponds to a separately trained model at that sequence length sufficiently resolves the apparent contradiction.



Concerns That Remain Outstanding

Evaluation breadth (ETuF, ArMp):
While the authors justify perplexity as appropriate for architectural studies, concerns remain that the evaluation does not convincingly demonstrate benefits beyond language modeling efficiency, particularly for reasoning-heavy or autoregressive settings.

Speedup and systems-level clarity (ETuF):
The rebuttal provides high-level clarification but does not fully resolve ambiguity around wall-clock measurements, prefill vs. decode behavior.

Scalability to modern LLM regimes (ArMp):
The argument that 100M–1B models are sufficient for proof-of-concept is reasonable, but the lack of evidence at 7B+ scale remains a concern for practical impact.

Short-sequence and downstream degradation (ArMp):
Although acknowledged as a known limitation and framed as orthogonal, the observed performance drops on short-sequence benchmarks remain unresolved empirically.

**Reviewer Scores:**

Reviewer jkHd
Likely to increase slightly to 4.
The added comparisons with NSA/MoA and the clearer articulation of projection-level sparsity directly address the reviewer’s strongest objections. However, concerns about novelty relative to prior work and limited evaluation likely persist.

Reviewer ETuF
Likely to remain at 2.
While some misunderstandings about scope and metrics were clarified, the reviewer’s core concerns, evaluation unsoundness, KV-cache handling, speedup measurement, and lack of reasoning benchmarks, are only partially addressed.

Reviewer ArMp
Likely to increase to 6.
The rebuttal satisfactorily addresses several concerns, even if limitations remain.

---

### Decision · Program_Chairs · 2026-01-26

Reject